# Instrumented Learning of Clothes Hanger Insertion

Author names anonimised for review[1]

*Abstract*—Large behaviour models have transformed the field of robotic manipulation, but prohibitive data requirements have thus far prevented a revolution similar to vision language models. We believe that instrumentation, i.e. sensor integration in objects, can provide invaluable state information and enable efficient, robust learning for robotic manipulation. In this paper, we study instrumented imitation learning for the task of clothes hanger insertion. Using 200 teleoperated demonstrations, we train and compare Diffusion Policies under multiple ways of leveraging instrumentation: as state input, via soft sensor estimation, as auxiliary prediction targets, and through vision backbone pretraining. Results show that incorporating instrumentation signals during training can improve success rates by up to 20 %pt over a vision-only baseline, without requiring sensors at deployment. These findings demonstrate that instrumentation can be effectively used as privileged information to guide policy learning, offering a practical route toward more sample-efficient imitation learning for complex robotic manipulation tasks. Datasets are available on Zenodo [link redacted], selected rollout videos on Google Drive.

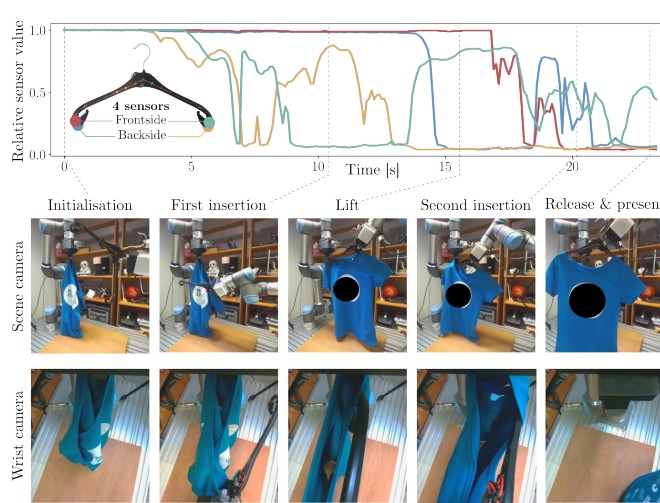

Fig. 1: Instrumented clothes hanger insertion.

## I. INTRODUCTION

Great expectations currently surround the development of large behaviour models for robotics [1]–[3] because of the recent success of vision-language models in image and text generation, or software-based environments in general. However, we are far from reaching the internet-scale data requirements to make them feasible [1], [2]. A popular way to achieve advanced manipulation skills for robots is to make them imitate expert behaviour through demonstrations, called imitation learning (IL) [4]. This has proven to be effective in teaching complex manipulation skills to robots [5]–[8], though learning even a single task in a controlled environment can take hundreds of demonstrations [5], [6].

In [9], a different approach from standard IL was used for raspberry harvesting: human demonstrators manipulated an instrumented, i.e. sensorised, strawberry phantom. Then, the parameters of a custom controller were tuned to match human behaviours as experienced by the strawberry. Instrumentation was also used in [10] for learning to solve a Rubik's cube with an anthropomorphic robot hand. Specifically, the sensor data provided accurate face angle observations of the Rubik's cube, adding to the state information needed to shape the reward function. However, when moving to a vision-only policy, the sensorised cube was discarded and replaced by a synthetically trained vision model: the sensorised cube was rather a stepping stone in development, than a critical influence in obtaining a functional policy. Similarly, **[self-citation redacted]** and **[self-citation redacted]** use sensors

for state estimation of cloth to enable robotic folding, but do not yet transfer to garments without sensors.

The fundamental idea of instrumentation is to use privileged information during the learning phase and then deploy without it, using only sensory input that is available in the field. This concept is also applied in works that learn robot behaviours in simulation [11], [12]. In [11], sim-to-real quadrupedal locomotion is achieved by first training a teacher policy that has access to privileged state information of the robot, allowing it to quickly achieve high performance. The teacher is then used to guide the learning of a student controller that only uses sensors available on the real robot. Similarly in [12], the student-teacher paradigm is used to learn in-hand reorientation. The teacher used the object orientation to quickly learn appropriate actions, after which the student was trained to match the teacher's outputs while relying on occluded visual observations of the objects. Instrumentation like in [9], [10] can be a way to obtain similar privileged information directly in the real world, potentially allowing for faster learning and/or more performant policies and avoiding the need for sim-to-real transfer, which already is a significant challenge on its own [2], [13].

Previously **[self-cite redacted]**, we presented a case study focusing on the task of clothes hanger insertion to uncover the potential of instrumented learning. We showed that (1) a black-box IL policy can recognise the importance of instrumentation signals without explicit guidance, (2) including instrumentation in the training data for an IL policy can increase both the policy's task awareness as well as overall performance, and (3) that rollouts from such an "expert" policy

*Funding acknowledgements anonimised for review.

[1]Author affiliations anonimised for review e.mail@anon.com

can be used as extra training data to enhance the performance of a "student" policy that cannot rely on instrumentation. This work reiterates on the clothes hanger insertion case study and evaluates additional approaches beyond dataset augmentation to leverage instrumentation for sample efficient imitation learning (section II-D): (1) Direct prediction of instrumentation signals via a "soft sensor", (2) including instrumentation signals in the policy predictions rather than observations, and (3) pretraining the vision encoder. These approaches are compared to two baselines: vision-only, and inclusion of instrumentation signals in the policy observation.

## II. MATERIALS & METHODS

### A. Task Description

We choose clothes hanger insertion as a testbed for instrumentation, because the (mostly) rigid clothes hanger lends itself to sensor integration, while the task as a whole is challenging and rarely addressed in the state-of-the-art of cloth manipulation [14]–[16]. Fig. 1 shows the progression of the insertion task. The following constraints apply to task initialisation: (1) Both the T-shirt and the clothes hanger are already held by the robot arms. (2) Only one T-shirt and one clothes hanger are used, always held in the same location. Once started, the robot inserts the first leg of the hanger in the open collar of the T-shirt, and lifts the clothes hanger until it is in a good position to perform the insertion of the second leg. After the second insertion, the T-shirt is released so that it hangs only from the clothes hanger.

### B. Hardware setup

*1) Instrumented Clothes Hanger:* A standard clothes hanger is instrumented by integration of four QRE1113GR reflective infrared (IR) range sensors on flexible printed circuit board strips, see Fig. 2. The QRE1113GR contains an IR light-emitting diode (LED), and an IR phototransistor (PT). When the clothes hanger is not covered, the IR light radiates away, but if the T-shirt covers it, the light reflects back to the PT, causing a large change in the photocurrent, as can be seen in the top curves of Fig. 1. For readout, we present an open-source[1], modular, wireless readout module **[name redacted]**. The readout module, as well as a 150 mAh LiPo battery, are placed in a 3D-printed holder and attached to the clothes hanger using velcro tape. Data is communicated to a workstation over Bluetooth Low Energy (BLE).

*2) Robot Setup:* The robot system is shown in Fig. 3a. It features two UR5e collaborative robot arms: the left holds the T-shirt, the right moves the clothes hanger. A Schunk EGK-40-MB-M-B gripper is mounted to each arm. The left robot arm has a RealSense D405 wrist camera, an externally mounted Zed 2i camera provides a view of the entire scene.

*3) Teleoperation Setup:* We use a Gello [17] arm (Fig. 3b) to control the six joint positions of the right UR5e, the left UR5e is static. The Gello trigger controls the opening of the gripper on the left UR5e so that the T-shirt can be released.

---

[1]**github link redacted**

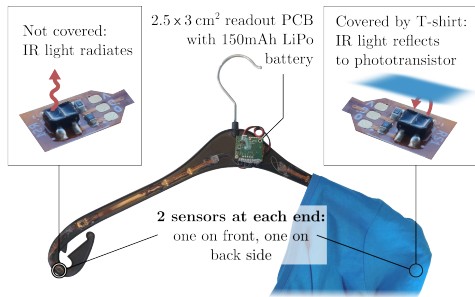

(a) Sensor integration and working principle.

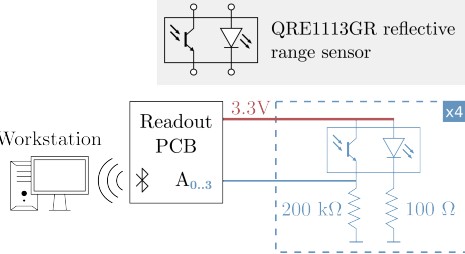

(b) Electrical schematic.

Fig. 2: Instrumented clothes hanger with range sensors, which respond strongly to covering of the clothes hanger.

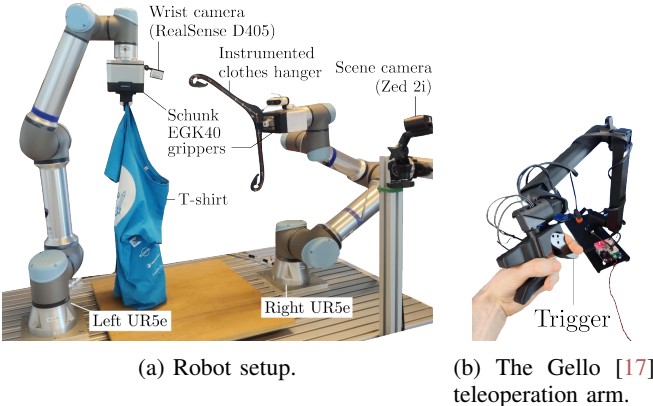

(a) Robot setup.  (b) The Gello [17] teleoperation arm.

Fig. 3: Experimental setup.

### C. Data Collection

The initial state of the T-shirt is randomised by drawing a line with uniform random angle and distance on the live wrist image, holding the second shoulder of the T-shirt along that direction at that distance, and releasing (see Fig. 4). A single episode (teleop demo or policy rollout) can take three forms:

**Type I:** Start position as depicted in Fig. 1 and 3a, with the right UR5e in a fixed home pose. The task is executed to completion.

**Type II:** The right arm starts from a "missed first insertion", i.e., the first leg of the clothes hanger is near to, but below the level of, the collar of the T-shirt. The task is executed to completion.

**Type III:** The right arm starts from a "missed first insertion", the episode ends right after the first insertion.

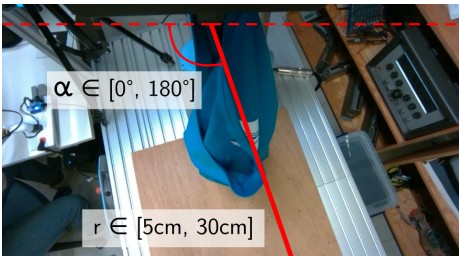

Fig. 4: Random initialisation of the T-shirt.

In total, 20 Type I, 150 Type II and 30 Type III demonstrations are collected, for a total of 200 demonstrations. This comprises 44 189 observations, which accounts for 1.2 h of data. Datasets are available on Zenodo [**link redacted**].

### D. Policy Architectures & Training

We use a Diffusion Policy (DP) [18] architecture with a ResNet18 vision backbone (with global average pooling instead of spatial softmax as in the original DP architecture [18], prior work indicates spatial softmax can hurt performance [19]) and a CNN-based noise prediction network. We set the hyperparameters to the default values from [18] unless specified otherwise. We evaluate five architectures, depicted in Fig. 5:

**Vision-only (Fig. 5a):** Standard DP architecture as baseline, the vision backbone encodes both camera images separately and concatenates the embeddings. The seven teleoperation degrees of freedom (six joints + gripper) form the "state", which is concatenated to the image embeddings. This vector is called the global conditioning vector and is the input to the diffusion model.

**Instrumentation in state (Fig. 5b):** The four sensor values from the instrumented clothes hanger are concatenated to the global conditioning vector. This gives the policy ground-truth information about the state of the clothes hanger, but also makes the policy dependent on this privileged information.

**Soft sensor (Fig. 5c):** The term "soft sensor" applies to algorithmically deriving low-dimensional, unobservable quantities from high-dimensional measurements [20]. Here, a separate ResNet18 is trained with a fully connected regression head to explicitly predict the instrumentation values given a wrist image or a scene image. For inference, we use the trained "Instrumentation in state"-network, but replace the ground truth instrumentation input with the predicted soft sensor values. This way, the "Instrumentation in state"-network becomes independent from the privileged instrumentation data.

**Instrumentation in action (Fig. 5d):** The clothes hanger sensor values are appended to the *action* vector, requiring the policy to explicitly predict the sensor on top of robot actions. This is akin to an auxiliary task in reinforcement learning [21].

**Pretrained backbone (Fig. 5e):** The soft sensor network (without regression head) is used as the initial

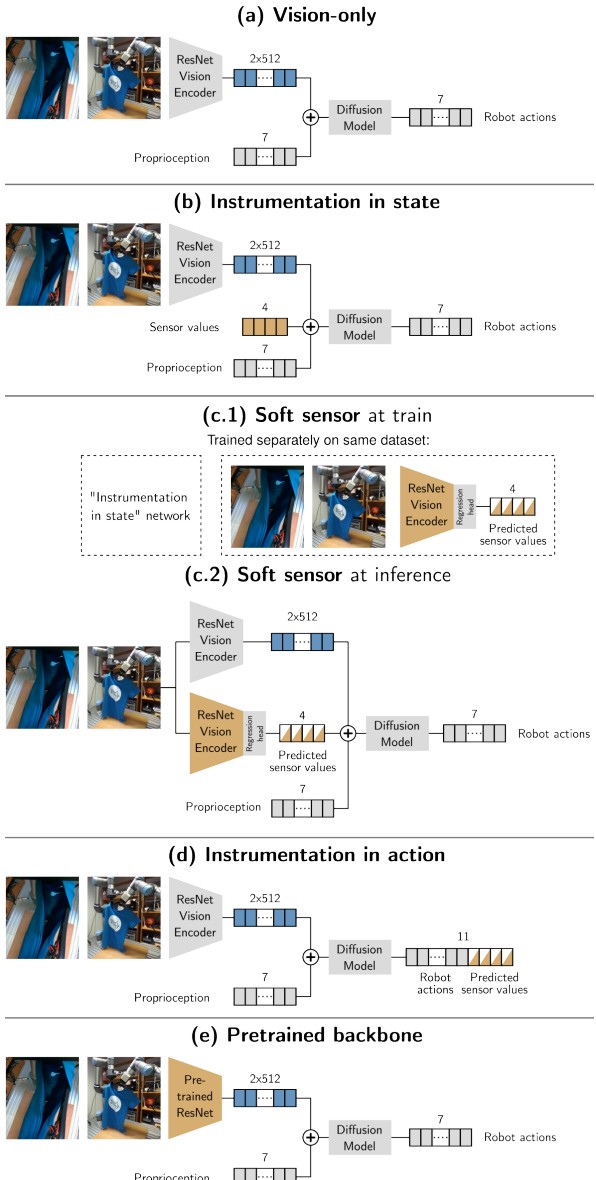

Fig. 5: Policy architecture diagrams (see section II-D).

backbone weights for training the "Vision-only" architecture.

Each DP predicts a chunk of 16 actions, of which 8 are executed before predicting another chunk. At 10 Hz, these 8 actions correspond to 0.8 s of wall-clock time. We train all Diffusion Policies for 300 000 steps with a batch size of 64 (434 epochs), which takes ~23 h on an NVIDIA H200 NVL, inference takes 28 ms on an NVIDIA RTX4090. For the soft sensor network, validation loss was found to saturate at around 15 epochs (33 150 steps with batch size 40), inference takes 2 ms on an NVIDIA RTX4090.

### E. Evaluation

All policy rollouts used to calculate success rates are of Type I. "Success" is defined as performing all stages depicted in Fig. 1 within a time limit of two minutes. Section III reports experimental success rates on 40 rollouts per policy,

and also provides a Bayesian interpretation to better represent the results [3]: task success is treated as a Bernoulli distribution with unknown parameter $p$, the probability of success, for which we can derive a probability density function $f_{p|\hat{p}}(p|\hat{p})$ given the experimentally determined success rate $\hat{p}$. Assuming a uniform distribution on $p$, it follows that $p|\hat{p} \sim \text{Beta}(N\hat{p}+1, N(1-\hat{p})+1)$ **[self-citation redacted]**, with $N$ the number of evaluation rollouts, i.e., 40.

## III. RESULTS & DISCUSSION

The end-to-end success rates with Bayesian estimated distributions are shown in Fig. 6, failure mode incidence is reported in Table I. We distinguish the following failure modes: **Timeout [1st insertion]:** First insertion not completed before 2 min timeout; **Pulled drop:** T-shirt pulled from between the fingertips during first insertion attempt; **Timeout [2nd insertion]:** Policy gets stuck right before the second insertion until timeout; **Collision [2nd insertion]:** Collision occurs between clothes hanger, gripper, and/or wrist camera during attempt of second insertion; **Drop [2nd insertion]:** T-shirt is released after an unsuccessful attempt of the second insertion and falls to the table.

The **vision-only** policy achieves a success rate of 65 %, with 10 out of 14 failures classified as **Drop [2nd insertion]**. This shows a lack of task awareness: the policy overfits on robot motions, neglecting to pay attention to the state of the clothes hanger and T-shirt. **Including instrumentation in the state** increases success rate by 10 %pt, and avoids the **Drop [2nd insertion]** failures altogether: the clothes hanger "knows" when it is properly inserted. This policy is, however, dependent on the privileged instrumentation data. Replacing the privileged data with **soft sensor** predictions decreases the success rate to 40 %: the soft sensor is noisy (NMSE of 0.08 on test set of 16 182 observations) compared to ground truth and confuses the DP. Both **"Instrumentation in action"** and **backbone pretraining** outperform the **vision-only** baseline by 20 %pt. Whereas the **vision-only** policy often did not pay attention to the clothes hanger before the second insertion, **making the policy predict the sensor values** focusses its attention to the state of the clothes hanger. Still, four **Drop [2nd insertion]** failures occur, but for one of these failed rollouts the drop only happened at the third attempt of the second insertion, and two successful rollouts of the **"Instrumentation in action"** policy recovered from an initially failed second insertion. The **vision-only** policy never attempted to recover from a failed second insertion. Lastly, **pretraining the backbone** induces a structure in the latent image space that captures information about the state of the clothes hanger, which is invaluable to the task of clothes hanger insertion, allowing the DP to find a local loss minimum that generalises much better to new observations. Interestingly, **including instrumentation in the state** does not achieve the highest success rate, even though the policy has direct access to privileged information and could therefore be expected to serve as an upper bound. We hypothesise that direct access to privileged information promotes shortcut learning [22], bypassing learning of gener-

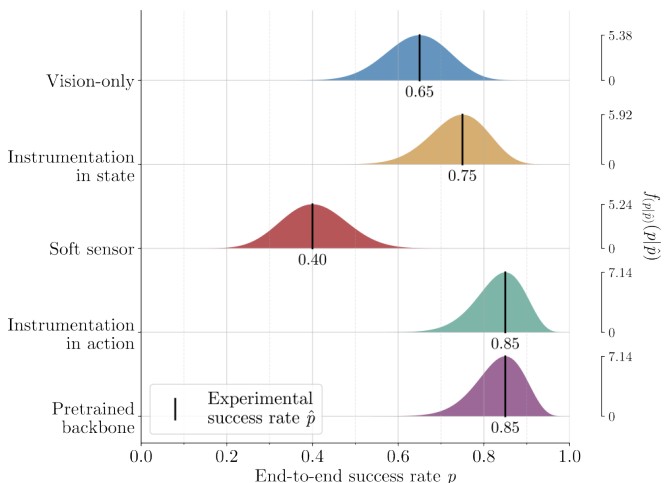

Fig. 6: Probability density functions (PDFs) of the end-to-end success rates for the trained architectures in Fig. 5.

TABLE I: Policy failure mode incidence on 40 rollouts (color scale normalised per row).

| Policy | #Failures | Timeout [1st insert] | Pulled drop | Timeout [2nd insert] | Collision [2nd insert] | Drop [2nd insert] |
|---|---|---|---|---|---|---|
| **Vision-only** | 14 | 2 | 1 | 0 | 1 | 10 |
| **Instr. in state** | 10 | 2 | 1 | 6 | 1 | 0 |
| **Soft sensor** | 24 | 8 | 0 | 10 | 4 | 2 |
| **Instr. in action** | 6 | 1 | 0 | 0 | 1 | 4 |
| **Pretrained backbone** | 6 | 3 | 0 | 2 | 0 | 1 |

alisable visual representations. Another possible explanation is causal confusion [23]: the policy waits for sensor cues to perform certain actions, but it does not realise it has to act in a certain way for those cues to emerge in the first place (cfr. six **Timeout [2nd insertion]** failures).

## IV. CONCLUSION

We have reiterated and improved upon clothes hanger insertion as a case study for instrumented learning. We proposed and evaluated three novel approaches to leverage privileged instrumentation data during training, while obtaining a policy that relies only on vision and proprioception. Including instrumentation signals as auxiliary prediction targets and pretraining the vision backbone were found to increase Diffusion Policy success rate by an impressive 20 %pt. These results manifest instrumented learning as a viable approach to increase sample efficiency for imitation learning. Extrapolating this, we envision that ground-truth object state information can be used in the real world to build more effective and robust manipulation policies. In future work, we will expand the case study to a larger variety of T-shirts and clothes hangers, and investigate other tasks where instrumentation could be beneficial.

## V. ACKNOWLEDGMENT

GPT-5 was used to write the first draft of the abstract. GPT-5 and Writefull were used sparingly throughout the manuscript to improve specific sentences and wordings.

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
