# OpenReview forum: "Instrumented Learning of Clothes Hanger Insertion"
_IEEE.org/ICRA/2026/Workshop/Manipulation_Robustness — ICRA 2026_

### Official Review · Reviewer_bi9T · 2026-05-06
**A good paper on instrumented learning with clothes hanger as a test case**

**Rating:** 8
**Confidence:** 5

**Review:**

Strength:
1) I really appreciate providing statistical analysis instead of purely success rate

2) Comprehensive comparison of different method of using the instrumented signal

3) Good writing, easy to follow and understand. Results are presented clearly, and analysis are in depth

Weakness:
1) Though I appreciate the work in general, robustness is not explicitly analyzed. A dedicated analysis of the robustness (aligning to the core of the workshop) of different methods would be nice. For example, varying different initial conditions

Overall: A well presented work that proposes and compares different methods to exploit instrumented signals, using cloth hanging as a testbed. Results show that both action prediction and encoder pertaining (using instrumented signal prediction as a pretraining task) are good ways. It would be very interesting and exciting to see if the conclusion applies and scales further.

---

### Decision · Program_Chairs · 2026-05-21

Accept